# Mortality of native and invasive ladybirds co-infected by ectoparasitic and entomopathogenic fungi

Danny Haelewaters[1,2,3], Thomas Hiller[4], Emily A. Kemp[5], Paul S. van Wielink[6], David I. Shapiro-Ilan[5], M. Catherine Aime[3], Oldřich Nedvěd[2], Donald H. Pfister[1] and Ted E. Cottrell[5]

[1] Farlow Reference Library and Herbarium of Cryptogamic Botany, Harvard University, Cambridge, MA, United States of America
[2] Faculty of Science, University of South Bohemia, České Budějovice, Czech Republic
[3] Department of Botany and Plant Pathology, Purdue University, West Lafayette, IN, United States of America
[4] Institute of Evolutionary Ecology and Conservation Genomics, University of Ulm, Ulm, Germany
[5] Agricultural Research Service, Southeastern Fruit and Tree Nut Research Laboratory, United States Department of Agriculture, Byron, GA, United States of America
[6] Natuurmuseum Brabant, Tilburg, The Netherlands

## ABSTRACT

*Harmonia axyridis* is an invasive alien ladybird in North America and Europe. Studies show that multiple natural enemies are using *Ha. axyridis* as a new host. However, thus far, no research has been undertaken to study the effects of simultaneous infection by multiple natural enemies on *Ha. axyridis*. We hypothesized that high thallus densities of the ectoparasitic fungus *Hesperomyces virescens* on a ladybird weaken the host's defenses, thereby making it more susceptible to infection by other natural enemies. We examined mortality of the North American-native *Olla v-nigrum* and *Ha. axyridis* co-infected with *He. virescens* and an entomopathogenic fungus—either *Beauveria bassiana* or *Metarhizium brunneum*. Laboratory assays revealed that *He. virescens*-infected *O. v-nigrum* individuals are more susceptible to entomopathogenic fungi, but *Ha. axyridis* does not suffer the same effects. This is in line with the enemy release hypothesis, which predicts that invasive alien species in new geographic areas experience reduced regulatory effects from natural enemies compared to native species. Considering our results, we can ask how *He. virescens* affects survival when confronted by other pathogens that previously had little impact on *Ha. axyridis*.

Corresponding author
Danny Haelewaters,
danny.haelewaters@gmail.com

## INTRODUCTION

In nature, hosts may be exploited by more than one natural enemy. These organisms interact with each other and with their hosts (*Furlong & Pell, 2005*). Complex interactions shape the population structure and dynamics of all organisms in the system. Natural enemies also compete with one another, and the impact on the host can be either synergistic, additive, or antagonistic (*Shapiro-Ilan, Bruck & Lacey, 2012*). These interactions can be manifested in various aspects of host fitness or mortality. For example, biological control

of *Drosophila suzukii* (Diptera, Drosophilidae), an important pest of fruit and berry crops, can be improved by treatments combining multiple natural enemies, which have an additive effect (*Renkema & Cuthbertson, 2018*). At the same time, dual infections (even if causing an increase in host mortality) may be deleterious to one or both pathogens in terms of pathogen growth, fecundity, or other fitness parameters.

*Harmonia axyridis* (Coleoptera, Coccinellidae), native to eastern Asia, has rapidly increased its global range and is now present on all continents except Antarctica (*Roy et al., 2016*; *Camacho-Cervantes, Ortega-Iturriaga & Del-Val, 2017*; *Hiller & Haelewaters, 2019*). Even though it has repeatedly been introduced for its beneficial properties as a biological control agent against aphid pests, its negative effects on native ladybird communities in invaded areas (*Koch, Venette & Hutchison, 2006*; *Honěk et al., 2016*; *Brown & Roy, 2018*) and on food production (*Koch, Venette & Hutchison, 2006*) have raised serious concerns since the early 2000s (*Roy et al., 2016*). It is now a model organism for studying invasive alien species (*Roy & Wajnberg, 2008*; *Brown et al., 2018*) and it has been listed in Europe as "one of the worst" invasive species (*Nentwig et al., 2018*). *Harmonia axyridis* is often reported as a host to several natural enemies. These include parasites (*Hesperomyces virescens*, *Coccipolipus hippodamiae*, *Parasitylenchus bifurcatus*), parasitoids (phorid and tachinid flies, *Dinocampus coccinellae*, *Homalotylus* spp., Tetrastichinae spp.), pathogens (bacteria, fungi, nematodes, protozoans), and predators (bugs, lacewings, ladybirds, and spiders) (*Garcés & Williams, 2004*; *Riddick, Cottrell & Kidd, 2008*; *Riddick, 2010*; *Harding et al., 2011*; *Raak-van den Berg et al., 2014*; *Haelewaters et al., 2017*; *Ceryngier et al., 2018*). Independent studies show that natural enemies of native ladybirds have recently employed *Ha. axyridis* as a new host, sometimes simultaneously (*Raak-van den Berg et al., 2014*; *Haelewaters et al., 2017*; *Ceryngier et al., 2018*; *Knapp et al., 2019*). Review of the effects of parasites, pathogens, and parasitoids of *Ha. axyridis* shows that they have only limited potential for controlling population densities of their host when acting alone (*Roy et al., 2008*; *Riddick, 2010*; *Haelewaters et al., 2017*; *Ceryngier et al., 2018*). However, thus far, no studies have focused on the effects of infections of multiple natural enemies on *Ha. axyridis*.

*Hesperomyces virescens* (Ascomycota, Laboulbeniomycetes, Laboulbeniales) is a common obligate ectoparasite of ladybirds (*Roy et al., 2016*; *Haelewaters et al., 2017*). Although known since 1891, it was shown only recently that *He. virescens* is in reality a complex of multiple host-specific species (*Haelewaters, De Kesel & Pfister, 2018*). Contrary to most multicellular fungi, *He. virescens* as well as other members of the order Laboulbeniales lack hyphae; instead they form 3-dimensional multicellular thalli by determinate growth (*Blackwell, Haelewaters & Pfister, in press*). Laboulbeniales, including *He. virescens*, cannot be grown in axenic culture and no asexual stages are known, which makes their study challenging (*Haelewaters, Blackwell & Pfister, 2021*). Given locally high prevalence of *He. virescens* on ladybird hosts (*Riddick & Cottrell, 2010*; *Haelewaters et al., 2017*) and the abundance of entomopathogenic fungal strains in the environment (*Roy & Cottrell, 2008*), we examined mortality of native and invasive *He. virescens*-infected ladybirds exposed to either *Beauveria bassiana* or *Metarhizium brunneum* (Ascomycota, Sordariomycetes, Hypocreales) (*sensu Cottrell & Shapiro-Ilan, 2003*; *Cottrell & Shapiro-Ilan, 2008*). Because *He. virescens* forms a branched, non-septate, rhizoidal haustorium (*Weir & Beakes, 1996*)

that penetrates the host's exoskeleton and makes contact with the body fluid for nutrient uptake, we hypothesized that high thallus densities with concomitant haustorial formation by *He. virescens* weaken host defenses, thus increasing the host's susceptibility to infection by other natural enemies. With this experiment, we assess how *He. virescens* affects ladybird survival when exposed to other natural enemies that alone have little impact on *Ha. axyridis* populations, and we compare results with a North American-native ladybird of similar body size, *Olla v-nigrum*. If *He. virescens*—on its own and in combination with other natural enemies—significantly impacts survival of the invasive ladybird but not the native one, then the results of this work could have consequences toward a pest management strategy to control infestations of vineyards and agroecosystems by *Ha. axyridis*.

## MATERIALS & METHODS

### Field collections and laboratory colonies

*Harmonia axyridis* and *Olla v-nigrum* ladybirds were collected for the purpose of establishing laboratory colonies of *Hesperomyces*-infected and non-infected ladybirds. Specimens were collected at overwintering sites at the 485-ha USDA-ARS, Southeastern Fruit and Tree Nut Research Laboratory, located in Byron, Georgia, USA (32.657792, -83.7383580). Sex and age of field-collected specimens were not determined to reduce dispersal of fungal propagules (*Cottrell & Riddick, 2012*). All specimens were brought to the laboratory and housed in individual Petri plates (10 cm diam.) with 1/3 of a piece of a cotton dental wick (Deerpack Products, LLC, Miami, Florida) drenched in water for hydration. Ladybirds were housed in environmental chambers at 25 $\pm$ 1 °C and photoperiod of 14:10 (L:D) h. Food was provided 3 $\times$ per week in the form of *Ephestia kuehniella* eggs (Lepidoptera, Pyralidae) and an artificial meat-based diet (Beneficial Insectary, Redding, California). *Olla v-nigrum* and *Ha. axyridis* ladybirds were maintained within the Petri plates for 14d and 21d (*Cottrell & Riddick, 2012*), respectively, at which time ladybirds were visually examined for presence of *Hesperomyces* using a dissecting microscope at 50$\times$ magnification. Eggs were harvested from ovipositing ladybirds and used to establish clean (free from fungal growth) laboratory-reared colonies of ladybirds with known age.

### Laboratory rearing of ladybirds

During examination for presence/absence of *Hesperomyces*, ladybirds were divided into two groups, infected and non-infected. Both groups of ladybirds were placed into plastic rearing containers of 19 $\times$ 13.5 $\times$ 9 cm (Pioneer Plastics, North Dixon, Kentucky), which were modified with two 3-cm diameter circular openings, one that was covered by 1 $\times$ 1 mm mesh to allow for air flow; and the second that was covered with a removable #7 rubber stopper to allow for feeding routinely as well as adding newly emerged laboratory-reared ladybirds. Routine maintenance included transferring ladybirds into fresh rearing containers at the end of each 7d period, which included nutrient supplementations of laboratory-reared yellow pecan aphids, *Monelliopsis pecanis* (Hemiptera, Aphididae).

The first laboratory generation of adults emerged about one month after placement in rearing containers. Emerging adults were placed into fresh rearing containers and stored

into a separate incubator (25 $\pm$ 1 °C, 14:10 (L:D) h) for 7 days. Similar to field-captured *O. v-nigrum* and *Ha. axyridis*, *M. pecanis* aphids were used as a diet augmentation. As the study progressed, we also incorporated black pecan aphids, *Melanocallis caryaefoliae* (Hemiptera, Aphididae), in the ladybird diet (3 × per week).

## Artificial transmissions of *Hesperomyces*

Not only did we need the ladybirds for our experiments to be of the same age, we also needed to artificially infect a subset of these ''clean'' laboratory-grown, adult ladybirds with *Hesperomyces virescens*. Exposure to *Hesperomyces* was conducted via tumbling of field-captured 'source' ladybirds (infected with *Hesperomyces*) with randomly selected laboratory-reared 'target' ladybirds (*Cottrell & Shapiro-Ilan, 2008*). A total of 25 target ladybirds were mixed with 5 *Hesperomyces*-infected source ladybirds in a 1.6 × 5.8 cm glass tube, which was placed on a hot-dog roller (Nostalgia Electrics, Green Bay, Wisconsin) for 5 min. This procedure was repeated for at least 160 target ladybirds of both species. We only performed intra-specific artificial transmissions of *Hesperomyces*, meaning from source *Ha. axyridis* to target *Ha. axyridis* and from source *O. v-nigrum* to target *O. v-nigrum*. Both *Hesperomyces*-exposed target ladybirds and clean (unexposed) ladybirds were fed a diet of *M. pecanis* aphids for 24 h. We did a second tumbling experiment using randomly selected emerged adults from the second cohort of laboratory-reared colonies. More tumbling experiments were performed to increase quantities of *Hesperomyces*-infected ladybirds, but source/target numbers were changed to 100/40.

To reduce competition for food, ladybirds from all laboratory colonies were transferred from the plastic rearing containers to 14-cm diameter Petri plates. Ladybirds were provided with water ad libitum, *E. kuehniella* eggs, and artificial meat-based diet. Finally, for assay preparation, the ladybirds were transferred back to clean 19 × 13.5 × 9 cm plastic rearing containers by species.

## Dual fungal infections assay

Within 24 h preceding the assay, 160 non-infected and 160 *Hesperomyces*-infected ladybirds of each species (*Ha. axyridis* and *O. v-nigrum*) were each placed into sterile test tubes, one individual per test tube. Test tubes were then closed with a sterile foam stopper to prevent ladybirds from escaping while allowing for air flow. Infected ladybirds were divided into categories according to numbers of thalli per specimen. Because the assay would assess potential interactions between fungal infections, we aimed at selecting heavily *Hesperomyces*-infected ladybirds; as a baseline, we only used specimens in our bioassays with 14 or more thalli each.

The assay started by pipetting a 1 mL of $2.5 \times 10^5$ conidia/mL suspension to each test tube (*Cottrell & Shapiro-Ilan, 2003*; *Cottrell & Shapiro-Ilan, 2008*). Treatments included native *B. bassiana* (native Bb), a commercial *B. bassiana* strain (GHA Bb; Mycotrol ES, Mycotech, Butte, Montana), *M. brunneum* strain F52 (Mb, isolated from a tortricid moth, Austria 1971; Novozymes, Franklinton, North Carolina), and double-distilled water (ddH$_2$O) as a control treatment. Ladybirds were submerged and swirled for 5 s, after which the suspension was removed again using a pipette and each ladybird was placed into a 6 cm-diameter Petri plate.
Any remaining droplets of excess suspension was removed by touching only the droplet with a Kimwipe tissue (Kimtech Science Brand, Kimberly-Clark Worldwide, Roswell, Georgia). Petri plates with treated ladybirds were placed into an incubator (25 $\pm$ 1 °C, 14:10 (L:D) h). Food and cotton rolls drenched in water were provided ad libitum, and Petri plates were replaced as needed in all treatments and replications simultaneously. Ladybirds were observed for mortality and entomopathogen-induced mycosis at day 14. During assay #1, we made daily observations for ladybird mortality and mycosis. Upon death of a given ladybird, ample water was added to the cotton roll to provide moisture for entomopathogen growth and Parafilm was applied around the Petri plate to prevent spreading of the fungus. Deaths of ladybirds and visual confirmations of mycosis were recorded.

We performed 8 different treatments for each ladybird species: (1) *He. virescens*-positive + native Bb, (2) *He. virescens*-positive + GHA Bb, (3) *He. virescens*-positive + Mb, (4) *He. virescens*-positive + ddH$_2$O (control), (5) *He. virescens*-negative + native Bb, (6) *He. virescens*-negative + GHA Bb, (7) *He. virescens*-negative + Mb, and (8) *He. virescens*-negative + ddH$_2$O (double control). In a single assay, we replicated every treatment 3 or 4 times. We performed the entire assay with all treatments and replicated 3 times, using 6–10 ladybirds for each treatment. Note that *M. brunneum* treatments were used only in assay #3 (Table S1). Over all assays done during this study, we used 1,289 specimens of ladybirds (667 *O. v-nigrum* and 622 *Ha. axyridis*) (Table S2).

### Statistical analyses

All statistical analyses were performed in the R language and open-access environment for statistical computing (version 3.5.0; *R Core Team, 2018*). We used generalized linear mixed models (function glmer(), R-package *lme4*; *Bates et al., 2015*) to analyze the effect of the different treatments (GHA Bb, native Bb, Mb) on the survival of *Ha. axyridis* and *O. v-nigrum* in relation to prior infection with *Hesperomyces*. We modeled the binary response variable survival (alive/dead) of each ladybird individual for both host species separately, and used *Hesperomyces* infection status as well as the interaction of *Hesperomyces* infection status with treatment as explaining variables. Further, to correct for variation within replicates and assays, we included the random effect of treatment nested in replicate nested in assay. We compared our candidate models to a respective Null-model using likelihood ratio tests and, furthermore, calculated pseudo $R^2$-values (function r2(), R package *sjstats*; *Lüdecke, 2018*) to evaluate model fit. To visualize the modeling results and obtained model estimates as forest plots, we used the function plot_model() implemented in the R package *sjstats* (*Lüdecke, 2018*). For assay #1, we further fitted Kaplan–Meier curves to daily mortality data and tested for significant differences in mortality between ladybird species using the function survfit() of the R package *survival* (*Therneau & Lumley, 2019*).

## RESULTS

Our candidate models for both host species *Ha. axyridis* and *O. v-nigrum* were significantly better at explaining survival relative to chance variation (Chi-squared test, $\chi^2 = 156.7$,

$P < 0.001$; $\chi^2 = 153.0$, $P < 0.001$, respectively). The overall model fit was high for both candidate models (*Ha. axyridis*: Nagelkerke's $R^2 = 0.40$; *O. v-nigrum*: Nagelkerke's $R^2 = 0.53$) suggesting the variance is well described by our applied models.

We found a significant negative effect on ladybird survival of the *M. brunneum* treatment on *He. virescens*-negative *Ha. axyridis* (Fig. 1, Table 1), whereas *B. bassiana* treatments did not affect the survival of *He. virescens*-negative individuals. Infection with *He. virescens* significantly affected *Ha. axyridis* survival over all treatments (Fig. 1). However, there was no additional effect detectable among co-infection treatments for *He. virescens*-positive ladybirds (Table 1). Each treatment applied to *O. v-nigrum* had a significantly negative effect on the survival for both *He. virescens*-negative and -positive ladybirds (Fig. 1, Table 1). Finally, we found an additional negative effect of all co-infection treatments on the survival of *He. virescens*-positive *O. v-nigrum* (Fig. 1, Table 1). These results suggest that there is no effect of dual infections on *Ha. axyridis*, whereas *O. v-nigrum* is highly affected by simultaneous exposure to both *He. virescens* and an entomopathogenic fungus. Percentages of ladybird mortality by treatment are also presented in tabulated form in Table S3.

When comparing the daily survival of *Ha. axyridis* and *O. v-nigrum*, no significant differences were found in *Hesperomyces*-positive only treatments (log rank test, $P = 0.4$). However, when co-infected *O. v-nigrum* showed a significantly lower survival compared to *Ha. axyridis* for GHA and native *B. bassiana* strains (log rank test, $P = 0.0014$ and $P < 0.001$, respectively). Figure 2 shows how survival is significantly different between the two ladybird species when co-infected with both *Hesperomyces* and *B. bassiana* (GHA and native).

## DISCUSSION

Research on the additive effects of multiple natural enemies on a given host is rare, likely because of the complexity involved in designing robust bioassays that include all partners of the system. Combining the natural enemies *Orius insidiosus* (Hemiptera, Anthocoridae) and *Heterorhabditis bacteriophora* (Rhabditida, Heterorhabditidae) resulted in the largest decline in larvae of *Drosophila suzukii* (*Renkema & Cuthbertson, 2018*). This fruit fly causes major economic losses to fruit crops in its invasive range, spanning North and South America and Europe (*Lee et al., 2011*). The addition of *O. insidiosus* resulted in 50% fewer *D. suzukii* larvae compared to treatment with only *H. bacteriophora*. *Plutella xylostella* (Lepidoptera, Plutellidae), an important cosmopolitan pest of brassicaceous crops, offers another example. This organism shows resistance to almost all chemical insecticides (*Sarfraz, Keddie & Dosdall, 2005*). *Pandora blunckii* and *Zoophthora radicans* (Zoopagomycota, Entomophthoromycetes, Entomophthorales) both infect *P. xylostella* in the field. In co-inoculation studies with *Pa. blunckii* and *Z. radicans* in *P. xylostella* larvae, larval cadavers (three days post-mortality) were most frequently found with conidia of a single entomopathogen, usually the one that had been inoculated first (prior "residency")—meaning that the other species was excluded (*Sandoval-Aguilar et al., 2015*). In general, the presence of competing species in the same host resulted in a decreased proportion of *P. xylostella* larvae that were infected compared to single inoculations.
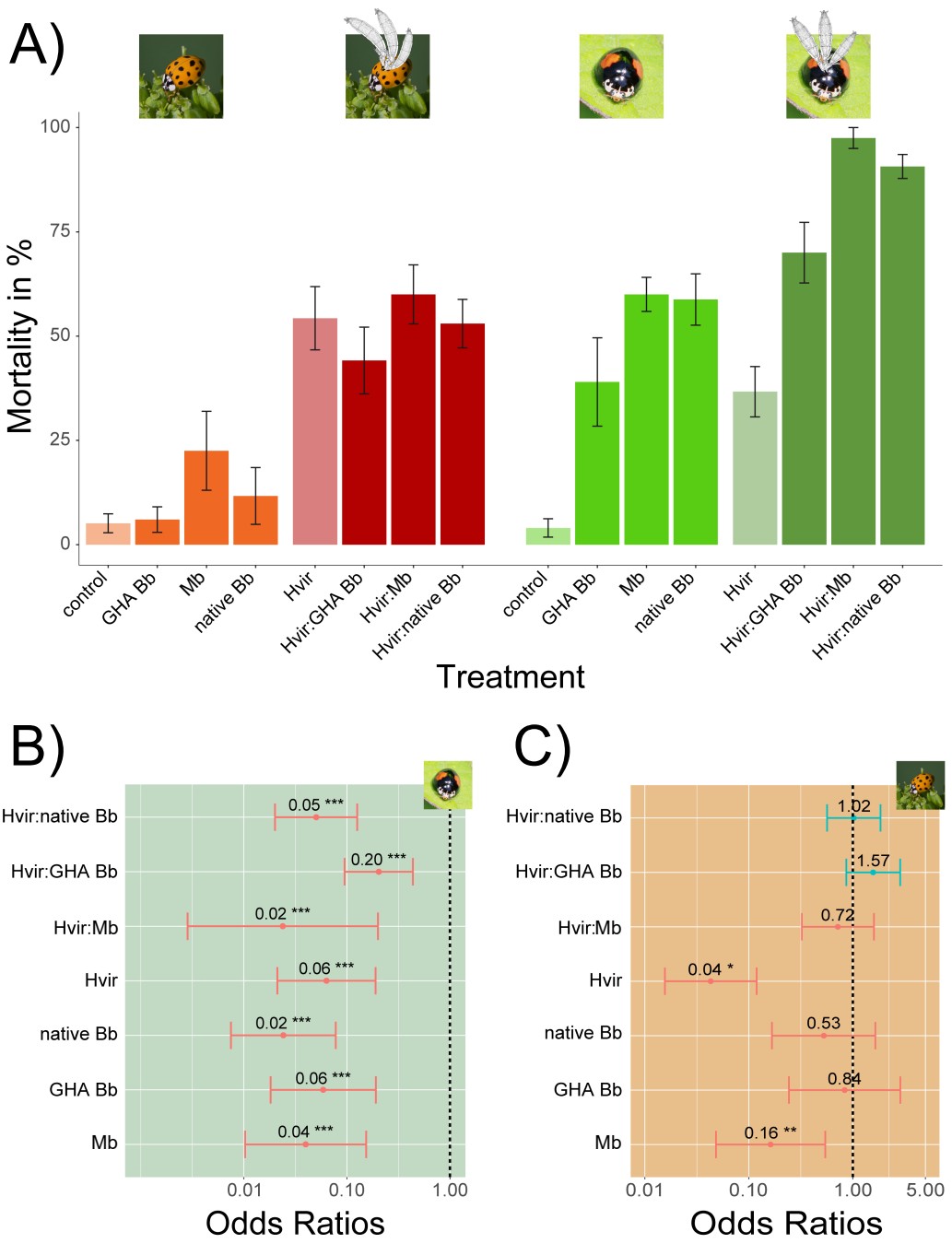

**Figure 1** **Results of treatment effects on native and invasive ladybirds.** (A) Percentages of ladybird mortality by treatment (left *Harmonia axyridis*, right *Olla v-nigrum*). (B, C) Forest plots illustrating the results of our modelling approach showing the treatment effects on survival of ladybirds (negative effect in red (odds ratio < 1), positive effect in blue (odds ratio > 1); ** *P* < 0.01, *** *P* < 0.001). (B) *Olla v-nigrum*. (C) *Harmonia axyridis*. Photo credits: *Olla v-nigrum*, Roberto Güller (Flickr); *Harmonia axyridis*, Andreas Trepte (http://www.photo-natur.net). Drawings of *Hesperomyces* thalli by André De Kesel. Abbreviations: Hvir, *Hesperomyces virescens*; GHA Bb, a commercial strain of *Beauveria bassiana*; native Bb, native *B. bassiana*; Mb, *Metarhizium brunneum*.

**Table 1  Results of our modeling approach for *Harmonia axyridis* and *Olla v-nigrum*.** Summary of parameters corresponding to *Hesperomyces virescens* infection and respective interactions with GHA *Beauveria bassiana*, native *B. bassiana*, and *Metarhizium brunneum*.

| | | Estimate | Std. Error | z value | P-value | |
|---|---|---|---|---|---|---|
| ***Harmonia axyridis*** | | | | | | |
| (Intercept) | | 2.977 | 0.563 | 5.284 | **<0.001** | *** |
| *He. virescens* infection | | −3.147 | 0.518 | −6.071 | **<0.001** | *** |
| *He. virescens*-negative | GHA Bb | −0.179 | 0.630 | −0.284 | 0.777 | |
| | Native Bb | −0.642 | 0.585 | −1.098 | 0.272 | |
| | Mb | −1.820 | 0.618 | −2.946 | **0.003** | ** |
| *He. virescens*-positive | GHA Bb | 0.454 | 0.305 | 1.487 | 0.137 | |
| | Native Bb | 0.024 | 0.302 | 0.080 | 0.936 | |
| | Mb | −0.330 | 0.406 | −0.811 | 0.417 | |
| ***Olla v-nigrum*** | | | | | | |
| (Intercept) | | 3.407 | 0.609 | 5.598 | **<0.001** | *** |
| *He. virescens* infection | | −2.757 | 0.560 | −4.925 | **<0.001** | *** |
| *He. virescens*-negative | GHA Bb | −2.831 | 0.599 | −4.728 | **<0.001** | *** |
| | Native Bb | −3.723 | 0.597 | −6.234 | **<0.001** | *** |
| | Mb | −3.222 | 0.689 | −4.676 | **<0.001** | *** |
| *He. virescens*-positive | GHA Bb | −1.591 | 0.390 | −4.084 | **<0.001** | *** |
| | Native Bb | −2.988 | 0.470 | −6.364 | **<0.001** | *** |
| | Mb | −3.734 | 1.084 | −3.444 | **<0.001** | *** |

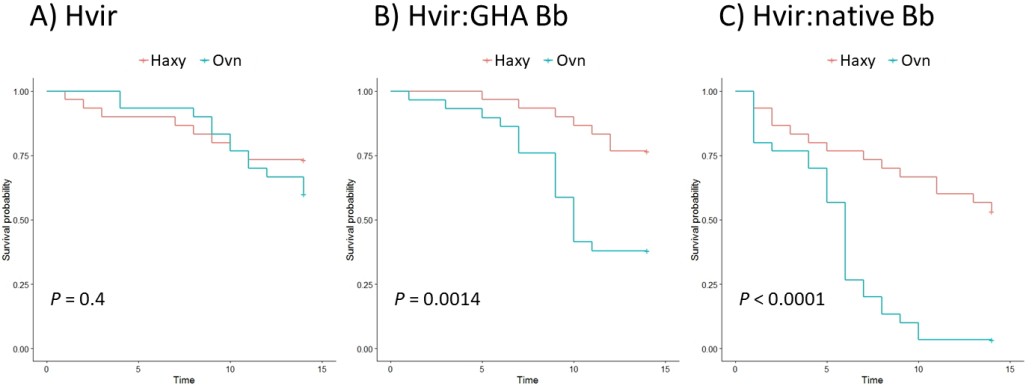

**Figure 2  Effect of *Hesperomyces*-infection and co-infection with *Hesperomyces* and *Beauveria bassiana* on the survival of ladybirds.** (A) Survival of *Hesperomyces*-positive ladybirds without dual infection. (B) *Hesperomyces*-positive ladybirds co-infected with GHA strain of *Beauveria bassiana*. (C) *Hesperomyces*-positive ladybirds infected with native *B. bassiana*. The survival of ladybirds is significantly different between *Harmonia axyridis* and *Olla v-nigrum* with the dual infection treatments.

Regarding *Ha. axyridis*, the following co-infections of natural enemies have been observed in nature: *He. virescens* + *Coccipolipus hippodamiae* mites (Acarina, Podapolipidae) in the USA, Austria, and the Netherlands (*Christian, 2001*; *Riddick, 2010*; *Raak-van den Berg et al., 2014*) and *He. virescens* + *Parasitylenchus bifurcatus* nematodes (Nematoda, Allantonematidae) in the Czech Republic, Germany, and the Netherlands

(*Raak-van den Berg et al., 2014*; *Haelewaters et al., 2017*; *Herz & Kleespies, 2012*). Given the status of *Ha. axyridis* as an invasive alien species, these findings demand a better understanding of interactions among the different natural enemies and their potential role in limiting populations of *Ha. axyridis*. To date, bioassays to determine mortality of ladybirds induced by infection by one or more natural enemies have not yet been performed. Likewise, bioassays including Laboulbeniales have only been carried out in one study (*Konrad et al., 2015*).

When we started this study, *He. virescens* was considered a single species with many ladybird hosts, potentially with strains that infect only a single species or closely related ones (*Cottrell & Riddick, 2012*). However, it was recently shown that *He. virescens* is a complex of multiple species, each with its own ladybird host (*Haelewaters, De Kesel & Pfister, 2018*), which calls for caution in reviewing reports from the extensive body of literature on *Hesperomyces* findings (summarized in *Haelewaters & De Kesel, 2017*). This also means that isolates of *He. virescens* from *Ha. axyridis* and *O. v-nigrum* in fact represent two different species of *Hesperomyces*. In other words, the experiments in the current study allow us to make comparisons between two host species, each with their own specific fungal parasite. Future experiments are needed to further disentangle these interactions. Even though horizontal transmission of *Hesperomyces* among ladybird species is rare (*Cottrell & Riddick, 2012*), we should try to infect *Ha. axyridis* and *O. v-nigrum* ladybirds with the species of *Hesperomyces* specific to *Olla* and *Harmonia*, respectively, perform bioassays, and compare mortality rates under different treatments with our current results. Analyzing interactions among natural enemies only make sense when the taxa considered represent single biological species.

We found a significant negative effect of *Hesperomyces*-only infection on the survival of both ladybird hosts (Fig. 1A, Table S3). Previous work has shown *Hesperomyces* infections to result in decreased mating frequency of female ladybirds, lower (male) survival rates in winter, and impeded sensing ability and flexibility of legs in heavily infected ladybirds (*Nalepa & Weir, 2007*; *Riddick, 2010*; *Haelewaters et al., 2017*). One study implicated parasitism by *He. virescens* as the cause of late summer mortality of *Chilocorus bipustulatus* ladybirds (*Kamburov, Nadel & Kenneth, 1967*) but this was later disputed based on controlled laboratory experiments (*Applebaum et al., 1971*). Our research is the first to explicitly link *Hesperomyces* infection with increased ladybird mortality.

Our findings on the effects of *Hesperomyces* on ladybird survival provided a unique opportunity for setting up dual infection assays—the first such experiments to be conducted on ladybirds. When first infected with *He. virescens* and then exposed to either *B. bassiana* or *M. brunneum*, *Ha. axyridis* mortality was not increased. This result was unexpected. We had hypothesized that *Ha. axyridis* with high thallus densities of *He. virescens* would have lowered host defenses against other pathogens. In contrast, the mechanism fostering low susceptibility of *Ha. axyridis* to entomopathogenic fungi (*Cottrell & Shapiro-Ilan, 2003*; *Knapp et al., 2019*) is not compromised by infection with *He. virescens*. Similarly, infection of *O. v-nigrum* by *He. virescens*-only increased mortality but—in contrast to *Ha. axyridis*—there was significantly higher mortality when co-infected by entomopathogenic fungi. Differential susceptibility to entomopathogenic fungi was reported by *Cottrell &*

*Shapiro-Ilan (2003)*, who showed that native *B. bassiana* was pathogenic to *O. v-nigrum* but not to *Ha. axyridis*. We confirm these results regarding the native strain but we also found the same differential pattern for the GHA strain of *B. bassiana*, whereas in the earlier study this strain was reported to be pathogenic to neither ladybird species (*Cottrell & Shapiro-Ilan, 2003*). It is perhaps surprising that we detect the GHA strain to be pathogenic to native ladybirds in contrast to the previous results, but ladybird populations may become more susceptible over time for various reasons and natural enemies also become better adapted (*Knapp et al., 2019*). We note that differential susceptibility has also been reported for entomopathogenic nematodes—again, *Ha. axyridis* was less susceptible compared to *O. v-nigrum* (*Shapiro-Ilan & Cottrell, 2005*).

In addition, our data are the first account of differential susceptibility to *M. brunneum* between the invasive *Ha. axyridis* and the native *O. v-nigrum*. Whereas infection with *M. brunneum* had a significantly negative effect on the survival of *He. virescens*-negative *Ha. axyridis*, this effect was not visible in the dual infection treatment. The infection with Laboulbeniales probably decreased the susceptibility of *Ha. axyridis* to infection by *M. brunneum*, similar to the findings of *Konrad et al. (2015)*. These authors found that *Laboulbenia*-infected *Lasius neglectus* ants (Hymenoptera, Formicidae) showed a decreased susceptibility to *M. brunneum*. This protection against *Metarhizium* was positively correlated with parasite load. Information on the parasite load of *He. virescens* on ladybirds in nature is nonexistent. In our bioassays, we selected ladybirds bearing 14 or more fungal thalli as *He. virescens*-positive specimens. Previous work from a long-term ATBI project in the Netherlands (*van Wielink, 2017*) points at an average of $19.8 \pm 4.9$ thalli and a maximum of 129 thalli per *Ha. axyridis* specimen ($n = 270$). No such data are available for *O. v-nigrum*. In other words, based on the available information, the artificial parasite load in our bioassays seems to closely mimic the natural conditions.

Our results provide direct support for the enemy release hypothesis (*Jeffries & Lawton, 1984*). This hypothesis is illustrative for the success of invasive alien species and stipulates that an invasive species in new geographic regions will experience reduced regulatory effects from natural enemies compared to native species, resulting in increased population growth of the invasive species (*Colautti et al., 2004*; *Roy et al., 2011*). However, invasions are dynamic and this escape-from-enemies could be lost as invasive species acquire new enemies over time (*Hokkanen & Pimentel, 1989*; *Schultheis, Berardi & Lau, 2015*; *Haelewaters et al., 2017*). Support for enemy release explaining the success of *Ha. axyridis* has come from two studies that reported decreased susceptibility of *Ha. axyridis* to entomopathogenic fungi (*Cottrell & Shapiro-Ilan, 2003*) and entomopathogenic nematodes (*Shapiro-Ilan & Cottrell, 2005*) compared to the native American ladybird species. Our work adds another level of complexity by the addition of a second natural enemy to the interactions. Again, we find differential susceptibility between the invasive and native ladybird species—with a reduced regulatory effect of the tested natural enemies on *Ha. axyridis*.

## CONCLUSIONS

In this paper, we show a negative effect of infection by *Hesperomyces virescens* on the survival of both *Harmonia axyridis* and *Olla v-nigrum* ladybirds (Fig. 1A). This is the first

study to unequivocally link *Hesperomyces* infection with increased host mortality and only the second to perform bioassays with hosts co-infected with Laboulbeniales and a second entomopathogenic fungus (*Konrad et al., 2015*). However, the susceptibility to a secondary entomopathogenic fungus was only elevated in the native American ladybird species (*O. v-nigrum*), whereas the globally invasive *Ha. axyridis* showed no significant increase in mortality when co-infected with either *Beauveria bassiana* or *Metarhizium brunneum* (Figs. 1 and 2). These findings are consistent with the enemy release hypothesis (*Jeffries & Lawton, 1984*) and highlight the difficulty in controlling this invasive alien species. Future studies are needed to elaborate population-specific effects on native and commercial strains of entomopathogenic fungi used in pest control.

## ACKNOWLEDGEMENTS

The authors would like to thank Merry Bacon and Chace Morrill (USDA, ARS, Byron, Georgia) for technical assistance.

### Funding
Danny Haelewaters was supported by the Georgia Entomological Society (GES) in the form of the 2015 Ph.D. Scholarship and by the Department of Organismic and Evolutionary Biology at Harvard University. The funders had no role in study design, data collection and analysis, decision to publish, or preparation of the manuscript.

### Grant Disclosures
The following grant information was disclosed by the authors:
Georgia Entomological Society (GES) in the form of the 2015 Ph.D. Scholarship.
Department of Organismic and Evolutionary Biology at Harvard University.

### Competing Interests
The authors declare there are no competing interests.

### Author Contributions
- Danny Haelewaters conceived and designed the experiments, performed the experiments, analyzed the data, prepared figures and/or tables, authored or reviewed drafts of the paper, and approved the final draft.
- Thomas Hiller analyzed the data, prepared figures and/or tables, authored or reviewed drafts of the paper, and approved the final draft.
- Emily A. Kemp performed the experiments, authored or reviewed drafts of the paper, and approved the final draft.
- Paul S. van Wielink and M. Catherine Aime analyzed the data, authored or reviewed drafts of the paper, and approved the final draft.
- David I. Shapiro-Ilan performed the experiments, authored or reviewed drafts of the paper, resources and supervision, and approved the final draft.

- Oldřich Nedvěd and Donald H. Pfister analyzed the data, authored or reviewed drafts of the paper, resources and supervision, and approved the final draft.
- Ted E. Cottrell conceived and designed the experiments, performed the experiments, analyzed the data, authored or reviewed drafts of the paper, resources and supervision, and approved the final draft.

## Data Availability

  Raw data is available in the Supplemental Tables.

## Supplemental Information

Supplemental information for this article can be found online at http://dx.doi.org/10.7717/peerj.10110#supplemental-information.

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
