# Peer review of "Mortality of native and invasive ladybirds co-infected by ectoparasitic and entomopathogenic fungi"

_PeerJ, doi:10.7717/peerj.10110_

## Round 0.1 · original submission · Minor Revisions

Reviewers have now revised the manuscript. Reviewers think the manuscript is robust, and the findings are interesting. I definitively agree with that. However, the material and methods section requires incorporating some observed aspects (pay special attention to reviewer 2). These points are all apparently easily approachable. In general, the manuscript involves minor changes. Please respond to each of the observations (point by point) made by the reviewers.

Reviewer 1 ·

Basic reporting

The English text was very clear and unambiguous throughout the manuscript. The literature references and background in the field was provided. The structure of the manuscript including figures and tables are professional. The raw data was shared in supplementary files. A hypothesis was presented and tested.

Experimental design

The research question was well-defined and meaningful. The research clearly fills a knowledge gap. The experimentation was suitably rigorous. The materials and methods were described in sufficient detail.

Validity of the findings

All data have been provided and they are adequately robust and appear statistically sound. Conclusions are well stated, linked to the hypothesis statement and supported by the results.

Additional comments

Dear Authors:
This is an interesting article seeking to experimentally manipulate pathogen densities to better understand why the invasive ladybird Harmonia axyridis has a competitive advantage over native ladybirds such as Olla v-nigrum in North America.

A few suggestions to improve the text:
1. Please add a few sentences in the Introduction or Materials and Methods to tell the readers that He. virescens is likely two separate stains in this study. One for Ha. axyridis, and one for O. v-nigrum. I see that you mentioned the two strains in the Discussion section.
2. In Figure 1, please change the x-axis label, maybe to Mean Percentage (+/- SE) of Mortality.
3. In Figure 2, I am hesitant to believe that the authors can honestly compare the infection rates between ladybird species, if He. virescens strains differ between ladybird species. Do the authors have any unpublished data to show that one strain is not more virulent than the other?
4. Line 426: Add italics to the sci. name.
5. Line 461, 469: BioControl

Reviewer 2 ·

Basic reporting

The theoretical framework must be more in line with the objective of the study. The information on lines 40-50 gives contradictory messages and it is unclear whether this study was designed from a fundamental ecological perspective, that is, to understand the impact of biotic interactions on the structure and dynamics of populations or if it was designed in a more applied perspective to contribute to finding the best solutions to control populations of Ha. axyridis. I wonder, is this period really necessary?

Literature, article structure, figures, tables and raw data are adequate.

Manuscript use a clear, unambiguous and correct text. We found nice profissional standards of courtesy and expression.

Experimental design

My major comments goes to experimental design.

The study adresses a timely topic of research and is of interest to ecologists, both those working in areas of biological invasion as well as those in the field of applied ecology. PeerJ is a suitable venue for its publication.

It is not clear why the procedure described in the lines 136-138 was carried out. Can I assume that this was only done to obtain more Hesperomyces infected ladybirds? Again, it is not clear why this procedure presented in lines 138-139 was carried out. Can I assume that this was done to obtain diferent Hesperomyces infectations levels?

I recomend that authors explain to what extent “14 or more thalli” exert a significant imumitarian response on ladybirds and that this response does not change differently to any value above 14 thalli and thus, does not alter susceptibility of hosts to aditional infections (see lines 151-153). This fact may explain the high variance in the percentages of ladybird mortality.

Authors should explain why they tested 1 mL of 2.5 × 105 conidia/mL suspension and not another concentration? Moreover, despite considering acceptable submerging ladybirds in a suspension with conidia, wouldn't it have been possible to use Potter's tower to spray ladybirds?

Why statistical comparisons of daily mortality between H. axyridis and O. nigr are performed without data of control tratments?

Validity of the findings

The study intends to test the effect effects of infections of multiple natural enemy on Ha. axyridis. Two merits/novelties can be highlighted from this study; i) the setting up of dual infection assays conducted on ladybirds and ii) the first account of differential susceptibility to M. anisopliae between the invasive Ha. axyridis and the native O. v-nigrum. The study is of interest to ecologists, both those working in areas of biological invasion as well as those in the field of applied ecology.

Additional comments

The study intends to test the effect effects of infections of multiple natural enemy on Ha. axyridis. Two merits/novelties can be highlighted from this study; i) the setting up of dual infection assays conducted on ladybirds and ii) the first account of differential susceptibility to M. anisopliae between the invasive Ha. axyridis and the native O. v-nigrum.

The experiments in the current study, contrary to what the authors advocated, does not fully allow to make comparisons of infections, and dual infections, between two host species. I believe that the study only allows the characterization of the effect of their two specific fungal parasite. This is clearly recognized by the authors when they refer to the need for additional experiments to disentangle the interactions. In view of this, it does not make sense to compare the daily mortality rate as shown in figure 2.
Minor comments:

Line 40: I recommend to replace “In nature and in agricultural ecosystems…” by “In nature…”.

Line 130-134: I recommend mentioning here the fact that the adult states have been mixed.

Reviewer 3 ·

Basic reporting

This manuscript aims to assess how mortality of two ladybird species, one native and one invasive, is mediated by the co-infection by entomopathogenic fungi. The question raised by the authors is very interesting and novel, dealing with the interactions among different parasitic fungi on the survival of two ladybird beetles. And the results are very interesting and meaningful. It is well written, with appropriate literature and good structure.
Nevertheless, I think that the introduction (background/context) should be modified in order to be more clear about 1) the previous information regarding these types of interactions; 2) the interaction between H. virescens and other fungi; 3) the fungi species selected and their origins; and 4) how the origin (of both ladybird beetles and fungi) could mediate the outcomes of these interactions (how co-evolutionary processes may mediate the interactions among these fungi with the ladybird beetle host species). Some of this information is provided very late in the manuscript, in the Discussion.
Very importantly, the hypothesis should include an explanation for why native and invasive ladybird species should respond differently to native or non-native fungi.

Experimental design

In general, the methods and statistical analyses are appropriate, and described with sufficient detail. A few issues that are not completely clear in the text are clarified when looking at the supplementary Tables.

Experiments include the establishment of laboratory–reared colonies of ladybirds, thus controlling for the initial parasitism state (absence of parasitic fungi), and age. It would be necessary to include a better justification of why using native B- bassiana and a commercial B. bassiana strain There´s nothing in the hypothesis dealing with fungi origin (native, exotic, natural or commercial strains). The authors should revise the hypothesis.

Validity of the findings

What it is not so clear to me is the Result section (see specific comments). Specially, I would like to see in the figures the results for the control treatments.

Hesperomyces virescens is a complex of multiple species, which is only reported in the Discussion. Because of this, in the Materials and methods I think it is necessary to be clearer about the origin of the Hesperomyces used in the trials. Did they come from H. axyridis? O. v-nigrum, both?

Specific comments:
L. 138-139: why the change in source/target numbers?
144: number of ladybirds per container? I suppose H. axiridis and O v-nigrum, infected and non-infected with H. virescens were reared in different containers. Yes?
L. 148: individually placed in the tubes?
L. 186-187: the controls are also treatments. How are they included in the analyses (in Table 1, the P-values are between each treatment and the corresponding control? There are no comparisons among all different treatments)?
L. 191-192: trials = assays? Pease, be consistent
L. 210. It is not clear to me how Table 1 confirm this result. It seems to me that it would be more appropriate citing Table 1 at the end of sentence in Line 211 (after “no additional effect detectable among co-infection treatments…”.
L. 330-331. I don´t see this result in Fig. 1 because, again, controls are not shown.
In addition, I would like to see the results of all eight treatments in Figure 1 (as in Table S3). Why it is not reported the Controls without H. virescens for H. axyridis and O. v-nigrum? (i.e., only 7 bars –and not 8- shown for each ladybird species). Is it that all results are the difference of these treatments with the corresponding control? Sorry I am not familiar with the concept “manifold of mortality”. Please, clarify.
Table 1 and 2 should be merged.
Fig. 2: An interesting result is that apparently the survival of O. v-nigrum (in presence of H. virescens) is more affected by the infection of native Bb than GHA Bb. Again, why the controls (survival of H. axyridis and O. v-nigrum without H. virescens) are not shown here?

Additional comments

My general comments are already stated in the previous sections

---

## Round 0.2 · Minor Revisions

We now have the second round of reviews from our reviewers. Overall, this second version almost completely satisfies the suggestions of reviewers, including this editor. I strongly recommend resubmitting a latest version as soon as possible with the points indicated by the reviewers.

Reviewer 1 ·

Basic reporting

Same as before (in my previous review)

Experimental design

Same as before (in my previous review)

Validity of the findings

Same as before (as in my previous review)

Additional comments

In the Intro. or M & M section, please indicate that two strains of the fungus (Hesperomyces virescens) are likely present in the field. Please add a sentence or two to reveal this important information to your readers.

In Figure 2, I believe it is difficult to compare infection rate between lady beetle species, if the fungus (H. virescens) exists as two distinct strains, one on Harmonia axyridis and the other on Olla v-nigrum. The Results should be slightly revised to address this problem.

Reviewer 3 ·

Basic reporting

Regarding the Introduction, I do not see major changes. Only one final paragraph was added and none of my suggestions or concerns were considered, and their arguments for not doing so are not convincing. I think the authors should modified the introduction (theoretical background/context) as requested before (reviewer 2 also pointed out that theoretical framework must be more in line with the objective of the study; in this case, the comparison between the invasive H. axyridis and the native Olla v-nigrum to co-infection). It is not adequate to mention O. v-nigrum just at the end of the Introduction, without any justification what it is pretended to test with the inclusion of this native species in the analyses.
The origins of the fungi should be mentioned in the Introduction. For H. virescens, Roy et al. (2011; Biocontrol) state that is probably native to North America or at least present there since 1931. See also Orlova-Bienkowskaja et al. (2018) Plos One. Are the other species native to Northamerica? If not, when they were introduced to the country? For how long have they been in contact with H. axyridis and O. v-nigrum in US? This information is very important when comparing the response of native and invasive ladybird species to the infection by pathogens.
I insist the hypothesis has to be revised, including what was expected when comparing native versus an exotic ladybird species because the manuscript, since the beginning (title), put special emphasis in how native versus exotic ladybird species respond to co-infection by ectoparasitic and entomopathogenic fungi.

Experimental design

No comment

Validity of the findings

No comment

Additional comments

Most of my suggestions to the original version of the manuscript were considered in this new version. Nevertheless, I think that the Introduction has to be modified according to my comments above.
I insist Tables 1 and 2 should be merged. They both address the same thing, but for the two species of coccinellids.
Figure 2 is incorrect. All three plots are the same (see Fig. 2 in the original). Also, I insist that controls must be included in this figure. This point was also raised by reviewer 2 and the authors' response is not satisfactory. The design includes controls without H. viriscens and these should be included in all analyzes.

---

## Round 0.3 · accepted · Accept

It has been a tough and complex review. This editor considers that although some points still deserve further support for some aspects, in its current version, the manuscript satisfies the requirements emphasized in this journal, which are based on delivering a constructive peer-review process based on methodological soundness rather than subjective considerations of impact or novelty and has therefore been accepted for publication.